# Unravelling the structure of glycosyl cations via cold-ion infrared spectroscopy

Eike Mucha[1,2], Mateusz Marianski [1,4], Fei-Fei Xu[3], Daniel A. Thomas [1], Gerard Meijer [1], Gert von Helden [1], Peter H. Seeberger[2,3] & Kevin Pagel[1,2]

Glycosyl cations are the key intermediates during the glycosylation reaction that covalently links building blocks during the synthetic assembly of carbohydrates. The exact structure of these ions remained elusive due to their transient and short-lived nature. Structural insights into the intermediate would improve our understanding of the reaction mechanism of glycosidic bond formation. Here, we report an in-depth structural analysis of glycosyl cations using a combination of cold-ion infrared spectroscopy and first-principles theory. Participating C2 protective groups form indeed a covalent bond with the anomeric carbon that leads to C1-bridged acetoxonium-type structures. The resulting bicyclic structure strongly distorts the ring, which leads to a unique conformation for each individual monosaccharide. This gain in mechanistic understanding fundamentally impacts glycosynthesis and will allow to tailor building blocks and reaction conditions in the future.

[1] Department of Molecular Physics, Fritz Haber Institute of the Max Planck Society, Faradayweg 4-6, 14195 Berlin, Germany. [2] Institute of Chemistry and Biochemistry, Freie Universität Berlin, Takustraße 3, 14195 Berlin, Germany. [3] Department of Biomolecular Systems, Max Planck Institute of Colloids and Interfaces, Am Mühlenberg 1, 14476 Potsdam, Germany. [4] Present address: Hunter College, The City University of New York, 695 Park Ave, New York, NY 10065, United States. These authors contributed equally: Eike Mucha, Mateusz Marianski. Correspondence and requests for materials should be addressed to P.H.S. (email: peter.seeberger@mpikg.mpg.de) or to K.P. (email: kevin.pagel@fu-berlin.de)

n the 1890s, Emil Fischer introduced the glycosylation reaction that became crucial in the chemical assembly of carbohydrates[1]. The reaction is believed to proceed under most reaction conditions via a key ionic species, the glycosyl oxocarbenium ion[2,3]. However, the transient nature of this short-lived reaction intermediate impeded its isolation and detailed structural characterization.

The glycosylation reaction follows, for the most part, a textbook $S_N1$ mechanism that proceeds via a carbocation intermediate. The cleavage of a leaving group leads to the formation of a planar carbocation intermediate that can be attacked by a nucleophile from either side to yield two stereoisomeric products. Carbohydrate synthesis greatly benefits from full stereochemical control during glycosidic bond formation. The stereoselective formation of *trans*-glycosidic linkages is most reliably achieved when a C2-participating neighboring group is employed[4,5]. C2 acyl groups such as 2-*O*-acetyl or 2-*O*-benzoyl interact with the anomeric carbon and promote the formation of 1,2-*trans*-glycosidic linkages. Structural details concerning the exact type of interaction, which could range between oxocarbenium-type and acetoxonium-type ions have remained elusive (Fig. 1). The stabilization of glycosyl cations in the condensed phase was previously demonstrated using superacids[6], but the results cannot be easily transposed to classical glycosylation conditions. The interaction of participating groups with the anomeric carbon of a glycosyl cation is expected to substantially influence the ring conformation and in turn impact the kinetics and stereochemical outcome of the reaction[7,8]. Reaction optimization of glycosylation is for the most part purely empirical[4] and a better understanding of the key intermediate is important to select the best building blocks and reaction conditions for glycan synthesis.

Mass spectrometry (MS) is a widely used tool to analyze complex samples by measuring the mass-to-charge ratio ($m/z$) of ions. Separations purely based on $m/z$, however, renders this technique inherently blind to the ions' internal structure. Spectroscopic techniques, on the other hand, probe molecular properties that depend on the spatial arrangement of atoms and, therefore, provide rich information about the underlying structure. Cryogenic ion infrared (IR) spectroscopy can resolve structural details of complex molecules and their aggregates such that even minute structural variations in isomeric oligosaccharides exhibit unique IR fingerprints, that allow for unambiguous assignments[9,10].

Here, the structures of three glycosyl cations of C2-acetylated and C3-, C4-, C6-methylated D-glucopyranose, D-mannopyranose, and D-galactopyranose (Supplementary Note 1) are determined in detail by cold-ion infrared spectroscopy. Helium nanodroplets are used as an ideal cryogenic matrix and resemble the environment of low dielectric constant solvents commonly used during glycosylations. The glycosyl donors to be structurally examined were specifically designed to decouple cation formation from other factors such as the influence of other participating or bulky protecting groups. Highly resolved IR spectra confirm the

covalent character of participating group interactions with the anomeric carbon and pinpoint the structural details of glycosyl cations such as ring puckering.

## Results

**Carbohydrate analysis using cryogenic IR-spectroscopy.** The experimental setup where $m/z$-selected ions generated by nanoelectrospray ionization (nESI) are accumulated in a hexapole ion trap and picked up by superfluid helium droplets traversing the trap was described previously[9,11]. The trapped ions are thermalized to the equilibrium temperature (0.4 K) of the helium nanodroplet that contains around $10^5$ helium atoms. Downstream of the instrument, the cryogenic ions inside the droplets are investigated with infrared radiation produced by the Fritz Haber Institute Free-electron Laser (FHI-FEL[12]). The resonant absorption of multiple IR photons causes helium evaporation and the ejection of bare ions that are detected by a time-of-flight (TOF) mass spectrometer. Finally, a highly resolved and reproducible IR spectrum, plotted as ion count at the TOF detector, is recorded.

Glycosyl cations are formed by in-source fragmentation of thioglycoside precursors (Supplementary Fig. 1). The recorded IR spectra exhibit well-resolved absorption bands between 900 and 1800 $cm^{-1}$ (Fig. 2). The vibrational modes contributing to this characteristic fingerprint region can be divided into three regions. The first two regions are dominated by complex C–O and C–C stretching modes below 1250 $cm^{-1}$ and low-intensity bending modes of C–OMe and C–H between 1250 and 1450 $cm^{-1}$. Both regions, however, are expected to yield coupled and anharmonic vibrations that do not provide enough characteristic bands for an unambiguous assignment of the cation's structure.

More direct evidence is found in the region above 1450 $cm^{-1}$, where C=O stretch vibrations of the acetyl group are expected. The exact position of the C=O vibration frequency strongly depends on the interaction with the anomeric center; in oxocarbenium-type structures, strong absorptions above 1600 $cm^{-1}$ indicate a free or weakly interacting carbonyl group, while the C1-bridged acetyl group in acetoxonium ions yields absorption bands below 1600 $cm^{-1}$. In the experimental spectrum, strong absorptions below 1600 $cm^{-1}$ suggest the formation of covalently bound acetoxonium-type ions.

**Conformational analysis using first-principles methods.** To elucidate the exact molecular structure, an extensive conformational search using genetic algorithms[13,14] and density-functional theory has been performed. In all cases, minima on the potential-energy surface readily divide into covalently bound acetoxonium-type species with a C=O–C1 bond distance below 1.6 Å and oxocarbenium-type species with C=O–C1 distances above 2.5 Å (Supplementary Fig. 2 & 3). The relative energetics of these two forms was further refined at the MP2 level of theory extrapolated to the complete basis set[15,16]. The calculations consistently

**Fig. 1** Glycosyl cation structures. Schematic representation of possible glycosyl cation structures of glycosylating agents containing a C2-participating group. After cleavage of the leaving group, the carbocation can adopt three hypothetical structures, which differ substantially in the interaction of the acetyl group with the anomeric carbon. The type of interaction affects the exact conformation of the ring pucker, which influences the kinetics of the subsequent nucleophilic attack and, as a result, the stereochemical outcome

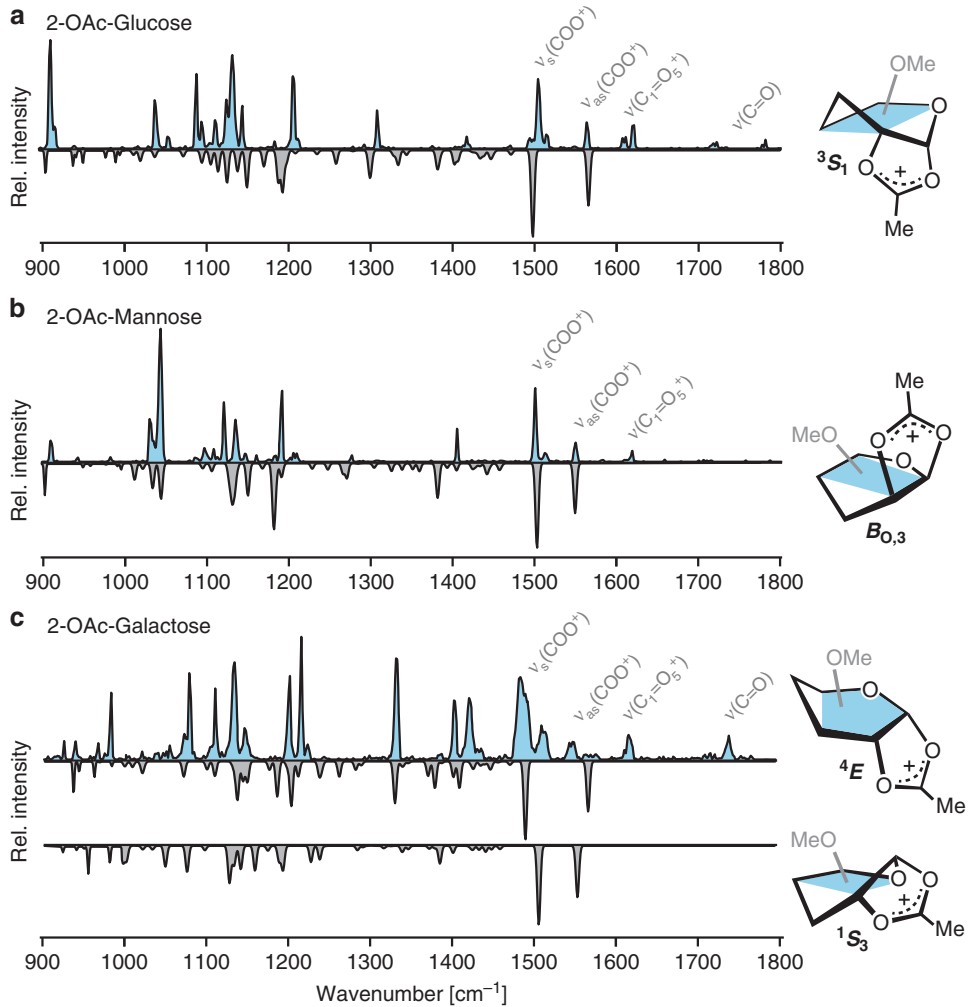

**Fig. 2** Infrared spectra of glycosyl cations reveal their conformations. Comparison of the experimental (blue) and theoretical (grey) IR fingerprinting region of **a** glucose, **b** mannose, and **c** galactose variants of glycosyl cations. The experimental spectra were recorded using cold-ion IR spectroscopy in helium nanodroplets and the theoretical spectra were derived using dispersion-corrected hybrid density-functional PBE0+D3 in 6-311+G(d,p) basis set. The highly diagnostic spectral region above 1450 cm$^{-1}$ is characterized by carbonyl vibrations as labeled; the fingerprint region below 1450 cm$^{-1}$ mostly contains coupled C–O, C–C, C–OMe, and C–H vibrations. On the right, the assigned ring puckers are schematically shown. The corresponding high-resolution structures are shown in Supplementary Figs. 4–6

predict that covalently bound acetoxonium-type ions are considerably more stable ($\Delta F_{\mathrm{harm}} > 6\,\mathrm{kcal\,mol^{-1}}$) than their oxocarbenium-type counterpart. A comparison of predicted IR spectra for the lowest-energy conformers (Fig. 2) with the experimental data shows a good agreement. The structures assigned to the glycosyl cations are shown schematically (Fig. 2). Detailed structures are shown in Supplementary Figs. 4–6.

**Spectroscopic analysis of cryogenic glycosyl cations**. The lowest-energy structure of the glucose variant is predicted to give characteristic symmetric and antisymmetric stretching modes associated with the covalently bound, C1-bridged acetyl group at 1497 and 1568 cm$^{-1}$, in good agreement with the experiment (Fig. 2a). The absorptions below 1500 cm$^{-1}$ are less indicative of the ions' structure, but the general agreement between experiment and theory supports the structural assignment. The bridged acetyl group aligns the six-member ring to adopt a $^3S_1$ ring pucker, where the positive charge at the acetyl carbon is stabilized by the axial OMe group at C4. In addition, weak absorptions appear above 1600 cm$^{-1}$. The lower energy bands around 1615 cm$^{-1}$ are associated with a [C1=O5$^+$] stretching mode of a free

oxocarbenium ion, whereas higher energy bands between 1700 and 1800 cm$^{-1}$ originate from unbound acetyl groups. The presence of these bands shows that a certain fraction of the ions adopt open structures.

For the epimeric mannose variant, the two diagnostic bands at 1500 and 1550 cm$^{-1}$, which again originate from covalently bound acetyl group, are well-reproduced by theory (Fig. 2b). The satisfying agreement for the remaining bands below 1500 cm$^{-1}$ further substantiate the structural assignment. This acetoxonium-type structure promotes the formation of a $B_{O,3}$ ring pucker. Here, the positive charge at the acetyl carbon is further stabilized by a spatially adjacent OMe group at C6. The sensitivity of the method towards ring puckering is demonstrated for this system as another low-energy candidate structure with a $^3H_4$ ring-pucker yields distinctly different vibrations for the acetyl group, which clearly discriminates against this ring pucker (Supplementary Fig. 7). Above 1600 cm$^{-1}$, only one weak absorption indicates that a small fraction of open-type structures coexist with the $B_{O,3}$-acetoxonium ions.

The galactosyl carbocation yields a more congested IR fingerprint (Fig. 2c). Additional intense bands between 1200

and 1450 cm$^{-1}$ appear, while the bands between 1450 and 1600 cm$^{-1}$ are broadened. Here, the lowest-energy structure adopts a $^4E$ ring pucker where the positive charge cannot be stabilized by the surrounding substituents. As a consequence, the covalent C–O–C1 bond is slightly elongated when compared to the respective bond in the other two cations (Supplementary Table 1). The predicted IR spectrum explains the identity of most of the additional bands in the region below 1450 cm$^{-1}$. Importantly, two bands at 1490 and 1570 cm$^{-1}$ associated with vibrations of the acetyl group align with theoretical results for the $^4E$ ring pucker. Only two small broader bands at 1510 and 1550 cm$^{-1}$ cannot be associated with any vibration of the lowest energy structure. However, their position matches very well with respective acetyl vibrations of a different low-energy structure which features an alternative $^1S_3$ ring pucker. Therefore, two structures with different ring puckers are likely to coexist for the galactose variant of the glycosylic cation. Again, two absorption bands above 1600 cm$^{-1}$ suggest that a certain fraction of open-type structures coexist with acetoxonium ions. In a conceptually similar work, Boltje et al. recently reported the direct characterization of glycosyl cations using IR multiple-photon dissociation (IRMPD) spectroscopy[17]. Although broader absorption bands were obtained due to the intrinsic thermal activation of the ions during the measurement, the covalent character of neighboring groups was confirmed. The enhanced spectral resolution obtained in this study, however, allows to confidently assign the different ring puckers predicted by theory.

## Discussion

In conclusion, we present a detailed structural characterization of glycosyl oxocarbenium ions, the key intermediate in glycosylation reactions. The combination of cold-ion IR spectroscopy and first-principles calculations provides evidence that cations with C2 participating protective groups adopt acetoxonium-type structures with a C1-bridged acetyl group. Distinct ring conformations are observed for each species. Diverse ring puckering influences the kinetics and stereochemistry of glycosidic bond formation and has to be taken into account when designing building blocks for glycan synthesis. Future experiments with building blocks containing C4 and C6 participating groups will shed light on the influence of remote participation on the stereochemical outcome of glycosylations.

## Methods

**Computational details**. The initial screening of the conformational space of 2-O-acetyl-D-glucopyranose, 2-O-acetyl-D-mannopyranose, and 2-O-acetyl-D-galactopyranose cations has been performed with a Fafoom genetic algorithm-based (GA) search tool[13,14] and FHI-aims full electron numerical atomic orbitals code[18]. Six rotatable bonds and a ring puckering have been selected as degrees of freedom. We performed 30 individual GA runs for each investigated carbocation. The settings of each GA run are shown in Supplementary Table 2. The local density-functional theory optimizations were carried out at dispersion-corrected PBE +vdW$^{TS}$[19,20] generalized gradient approximation level and in light basis set settings. Number of individual DFT optimizations are shown in Supplementary Table 3. In the next step, all structures for each carbocation were merged and clustered using RMSD distance matrix. The tight RMSD = 0.1 Å criterium between heavy atoms was selected to judge structural similarity. The RMSD calculations were performed in mdtraj python module[21], while hierarchical clustering was done with scipy module. The number of resulting unique structures for each cation is shown in Supplementary Table 3. For each structure, we performed single point energy evaluation at many body dispersion-corrected hybrid PBE0+MBD[22,23] level of theory and tight basis set settings in FHI-aims. Next, the distance between anomeric carbon C1 and the acetyl oxygen was measured for each structure and the relative energy as a function of this distance was plotted (Supplementary Fig. 2). In addition, the Cremer–Pople coordinates[24,25] of the ring pucker of each conformer was measured and a respective ring-pucker assigned.

The structures clearly separate into two distinct regimes—the covalently bound acetoxonium-type cation (C=O–C1 distance below 2.0 Å) and oxocarbenium-type cation (C=O–C1 distance above 2.0 Å). We selected multiple lowest energy structures from both regimes (number of selected conformers shown in

Supplementary Table 3) and performed geometry optimization at PBE0+D3/6-311 +G(d,p) level of theory[26] with default convergence criteria in Gaussian 09, Rev D.01[27]. After each optimization, we extract the C=O–C1 distance and ring pucker again and performed a frequency analysis within harmonic approximation. The presented IR spectra are normalized and scaled by 0.965 factor. The exemplary IR spectra for other conformers are shown in Supplementary Figs. 7–9.

Finally, the energy of each conformation was calculated at Resolution-of-Identity[15] MP2 level of theory, extrapolated to the complete basis set in ORCA program[28]. The extrapolation was done using two-point extrapolation[16] with def2-TZVPP and def2-QZVPP basis sets and auxiliary def2-QZVPP/C basis set for RI. We observed before that the RI yields virtually identical energies to those from MP2 calculations for monosaccharides[14]. Grid5 settings and tight SCF convergence were also requested. Finally, the conformational energies were augmented with free energy contributions from harmonic vibrational calculations performed at the DFT level.

## Data availability

All data is available from the authors upon reasonable request.

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

## Acknowledgements

The authors gratefully acknowledge generous funding by the Max-Planck-Society and the expertise of Sandy Gewinner and Dr. Wieland Schöllkopf of the FHI free electron laser. D.A.T. acknowledges support from the Alexander von Humboldt Foundation. M.M. is grateful to Matthias Scheffler (FHI Berlin) for support of the computational work.

## Author contributions

P.H.S. and K.P. designed and directed the research. P.H.S., K.P., G.v.H., and G.M. supervised the project. F.X. conducted chemical synthesis. E.M. and D.A.T. acquired experimental data. M.M. performed quantum chemical calculations. All authors wrote the manuscript, analyzed data, and interpreted the results.

## Additional information

**Competing interests:** The authors declare no competing interests.

