## [Peer Review File · Nature Communications]

Reviewers' comments:

Reviewer #1 (Remarks to the Author):

The authors report the structure of the glycosyl cations of Glucose, Galactose and Mannose, as identified by a combination of quantum chemistry and IR ion spectroscopy. Unravelling such structures is essential to the understanding of one of the most prominent biochemical reaction: glycosylation. As such, this work is not only of interest to the synthetic chemist, but also to the widest readership. The methodology is elegant and the paper is very pleasant to read.

General comment: The main claim of the authors is that this work constitute the first detailed characterization of glycosyl oxocarbenium ions, and that acetoxonium structures are formed (in the case of the 2-OAc species studied here). Although cryogenic spectroscopy is indeed used for the first time to address glycosyl cations, Boltje et al. have just released similar findings using a very similar approach (DOI: 10.1021/jacs.8b01236). This work must be cited and discussed. The authors may also cite several theoretical works by Bythell, and ion spectroscopy of B ions by Flitsch (DOI: 10.1021/acs.analchem.6b04998).

Specific comments:

- 1) The spectra shown in Fig. 2 are of excellent quality and convincingly show the presence of acetoxonium structures. Oxocarbenium structures however cannot be discarded based on the data shown. Indeed their spectroscopic diagnostic is $>1600\text{ cm}^{-1}$, and data in this range are not shown. Such diagnostic bands are visible in Figure S7 and the authors conclude that only a small fraction of oxocarbenium is present. It seems to me that both oxocarbenium and acetoxonium diagnostic bands have similar intensities. How do the authors evaluate the relative populations of structures ?
- 2) in Figures S5, S6 and S7, blue and black marks are used to represent calculated structures. Do they correspond to acetoxonium and oxocarbenium forms ? it would be helpful to specify it in the captions. If I interpret these figures correctly, structures B and C in Fig. S7 are oxocarbenium structures (the molecular drawing for C is difficult to see). If it is the case, I'm puzzled by the absence of the C=O mode above 1600 cm^{-1} in structure C. Could the authors elaborate ?
- 3) Finally, some marks could be added in Fig. 2 to indicate the nature of the most important diagnostic modes and ease the interpretation of the figure

Reviewer #2 (Remarks to the Author):

This work addresses a subject of considerable importance and of interest to the scientific community -- that is, identifying key intermediates in the chemical synthesis of carbohydrates. Carbohydrate synthesis is extremely challenging, and understanding the structure of intermediates can help guide the development of synthetic schemes.

The authors seek to be able to identify key intermediates using cold-ion infrared spectroscopy. While this is not a new technique, the authors have been pioneers in developing it and are leaders in the field. While others measure IR spectra of species in liquid helium droplets, no one else that I know has managed to do it with ions generated by electrospray. Their approach seems extremely well suited to the goal of this work.

The authors claim to have determined the structure of these intermediates by comparing measured IR spectra with the results of quantum chemical calculations. While I found their arguments somewhat convincing, there are a few issues worth mentioning that the authors may want to address:

- (1) The authors state on line 114 that the most direct evidence for the structure is found in the

region above 1450 cm⁻¹ where the CO stretch vibrations of the acetyl group are expected. In light of this, I found it somewhat surprising that they only show the spectra in this region up to 1600 cm⁻¹. It is well known that CO stretch vibrations can be substantially higher than this, and one is given the impression that they are perhaps trying to hide something. Indeed, in the supplementary information they show spectra up to 1800 cm⁻¹, and there seems to be additional bands in this region. This should be shown in the main text of the paper rather than stashed away in the SI. They also don't show the low frequency part of the spectrum below 1000 cm⁻¹. I don't understand why they don't show the entire spectrum in the main text.

(2) On line 130 they statement that experiment and theory show "remarkable agreement". It is not clear on what basis they say this. Looking at the SI in figure S7, I would say that for the spectra below 1450 cm⁻¹, none of the calculated spectra give better agreement than the other, unless one puts all the weight on the peak just above 900 cm⁻¹. I agree with the authors that the most direct evidence is provided by the spectral region above 1450 cm⁻¹.

(3) Regarding this higher frequency region, first of all, on line 135, I believe the authors mean to quote the calculated wavenumber, but the 1505 cm⁻¹ that they cite is clearly the experiment and not the theory. Other than this, in the caption of figure S7 where they show the entire spectrum, they state that minor bands above 1600 cm⁻¹ may stem from a small fraction of oxocarbenium-type ions. My question is, how do they know that this is only a small fraction? The peak around 1620 cm⁻¹, which seems to be on associated with an open structure, is almost as intense as the one at 1560 cm⁻¹. They need to justify the argument that this is only a minor amount. Moreover, they should do this in the main text and not in the caption of a figure in the SI.

(4) There is mistake on line 164. It should say Figure S8.

In general, I believe that the authors have probably gotten the assignments right, but they need to tighten the arguments for the comparison between experiment and theory. Otherwise, one could have simply done calculations and drawn conclusions about the structure of the key intermediates from theory alone. Once they tighten this up, I believe that the paper should be suitable for publication.

Reviewer #3 (Remarks to the Author):

The manuscript by Mucha et al presents the structural characterization of an important reaction intermediate in glycosylation via well-resolved, cold ion vibrational spectroscopy. This work is a timely demonstration of the power of cryogenic ion IR spectroscopy in accessing and characterizing directly relevant reaction intermediates. The results should be of interest to a wide audience.

However, an IRMPD study of the same cation intermediate by a Radboud University/FELIX team, just appeared in JACS. [J. Am. Chem. Soc. 2018, 140, 6034–6038] While the data presented in the Mucha et al. manuscript is of much higher quality, the JACS paper finds similar structures and reach similar conclusions. I think that this remove much of the "the first in-depth structural analysis of glycosyl cations" claimed by this manuscript and the novelty necessary for publication in Nature communications.

In addition I have the following comments about the manuscript.

1) The authors should consider linking the structure(s) determined in this study to the reactions involving these intermediates to make the results presented here more impactful to a wider readership. For example, what are the direct implications of the determined structures on the reaction stereochemistry, etc.

2) The crucial identifiers of the cation structure are the C-O/C=O stretch frequencies, and as such, the experimental IR spectra in Fig 2 should extend the frequency range to 1800 cm⁻¹ as shown in the SI. For most species, there are significant activity above 1600 cm⁻¹, possibly indicating presence of minor isomers of oxocarbenium nature. I don't think that this should be hidden in supplementary, as their presence would indicate the fluxional nature of such structures which may influence the reactivity. The ability of the experimental approach to capture higher energy isomers is particularly relevant when dealing with reaction pathways, where the lowest energy structures may actually be less interesting.

3) The fit of the calculated spectra in the finger print region is quite disappointing. In the JACS study, calculations at the B3LYP level appear to produce spectrum that match better with the experiment in the 800-1400 cm⁻¹ region. In any case, this make it difficult to assign a particular ring structure from the spectrum itself, and this assignment is mostly based on the calculated relative energy. If such assignment is to be made, a higher level calculation should be performed to confirm the energetic ordering.

Minor:

On line 135, the calculated CO stretch of glucose variant is listed as 1505 and 1560 cm⁻¹. But the calculated spectrum in fig. 2A appears to have the CO stretch below 1500 cm⁻¹. Authors may need to double check their values.

Response to reviewer's comments:

Reviewer #1:

The authors report the structure of the glycosyl cations of Glucose, Galactose and Mannose, as identified by a combination of quantum chemistry and IR ion spectroscopy. Unravelling such structures is essential to the understanding of one of the most prominent biochemical reaction: glycosylation. As such, this work is not only of interest to the synthetic chemist, but also to the widest readership. The methodology is elegant and the paper is very pleasant to read.

General comment: The main claim of the authors is that this work constitute the first detailed characterization of glycosyl oxocarbenium ions, and that acetoxonium structures are formed (in the case of the 2-OAc species studied here). Although cryogenic spectroscopy is indeed used for the first time to address glycosyl cations, Boltje et al. have just released similar findings using a very similar approach (DOI: 10.1021/jacs.8b01236). This work must be cited and discussed. The authors may also cite several theoretical works by Bythell, and ion spectroscopy of B ions by Flitsch (DOI: 10.1021/acs.analchem.6b04998).

We fully agree with the reviewer's comment on the recent work of Boltje et al. that came out shortly after we submitted this manuscript. Needless to say, we have added a paragraph to acknowledge and discuss their work. We also adjusted our statement on the "first" structural characterization of glycosyl cations.

Specific comments:

1) *The spectra shown in Fig. 2 are of excellent quality and convincingly show the presence of acetoxonium structures. Oxocarbenium structures however cannot be discarded based on the data shown. Indeed their spectroscopic diagnostic is $>1600\text{ cm}^{-1}$, and data in this range are not shown. Such diagnostic bands are visible in Figure S7 and the authors conclude that only a small fraction of oxocarbenium is present. It seems to me that both oxocarbenium and acetoxonium diagnostic bands have similar intensities. How do the authors evaluate the relative populations of structures ?*

We agree with the reviewer and extended the spectral range in Figure 2 from 900 to 1800 cm^{-1} to include the absorption bands above 1600 cm^{-1} and discuss them in the main text. Indeed, contributions of open-type structures are indicated by these absorptions and it is not easy to quantify the relative populations. For open-type structures, the C=O-stretch vibration is expected to give a high intensity absorption band between 1600 and 1800 cm^{-1} . For the mannose-variant, only one minor absorption is found above 1600 cm^{-1} and comparing its relative intensity to the bands at 1503 and 1550 cm^{-1} leads to the conclusion, that only a minor fraction of open-type structures is present. For the glucose and galactose variants, the case is indeed not too clear. We changed the text accordingly.

2) in Figures S5, S6 and S7, blue and black marks are used to represent calculated structures. Do they correspond to acetoxonium and oxocarbenium forms ? it would be helpful to specify it in the captions. If I interpret these figures correctly, structures B and C in Fig. S7 are oxocarbenium structures (the molecular drawing for C is difficult to see). If it is the case, I'm puzzled by the absence of the C=O mode above 1600 cm⁻¹ in structure C. Could the authors elaborate ?

The reviewer reads the figure correctly; the blue and black markers do indeed correspond to open- and closed type structures. We changed the figure captions to avoid ambiguity.

In Figures S7, S8 and S9, candidate structures B and C are oxocarbenium-type. The C=O modes are not absent, as theory predicts them around 1790 cm⁻¹, as shown in each figure. The band around 1600 cm⁻¹ present in structure B belongs to [C1=O5+] stretching mode within the vibration. In structure C, this vibration disappears due to binding of methoxy oxygen O6 to the anomeric carbon (distance 1.561 Å).

3) Finally, some marks could be added in Fig. 2 to indicate the nature of the most important diagnostic modes and ease the interpretation of the figure

We agree with the reviewer and added band assignments to highlight the most diagnostic vibrations.

Reviewer #2

This work addresses a subject of considerable importance and of interest to the scientific community -- that is, identifying key intermediates in the chemical synthesis of carbohydrates. Carbohydrate synthesis is extremely challenging, and understanding the structure of intermediates can help guide the development of synthetic schemes.

The authors seek to be able to identify key intermediates using cold-ion infrared spectroscopy. While this is not a new technique, the authors have been pioneers in developing it and are leaders in the field. While others measure IR spectra of species in liquid helium droplets, no one else that I know has managed to do it with ions generated by electrospray. Their approach seems extremely well suited to the goal of this work.

The authors claim to have determined the structure of these intermediates by comparing measured IR spectra with the results of quantum chemical calculations. While I found their arguments somewhat convincing, there are a few issues worth mentioning that the authors may want to address:

(1) The authors state on line 114 that the most direct evidence for the structure is found in the region above 1450 cm^{-1} where the CO stretch vibrations of the acetyl group are expected. In light of this, I found it somewhat surprising that they only show the spectra in this region up to 1600 cm^{-1} . It is well known that CO stretch vibrations can be substantially higher than this, and one is given the impression that they are perhaps trying to hide something. Indeed, in the supplementary information they show spectra up to 1800 cm^{-1} , and there seems to be additional bands in this region. This should be shown in the main text of the paper rather than stashed away in the SI. They also don't show the low frequency part of the spectrum below 1000 cm^{-1} . I don't understand why they don't show the entire spectrum in the main text.

We thank the reviewer for this helpful comment, which is in-line with the remarks of the other reviewers, and fully agree. Whereas our intention was to emphasize the characteristic region of the spectra, we now see that it can give a false impression. We now show the complete IR spectra in the main text and expanded the discussion accordingly (see also comment (1) of referee 1 above).

(2) On line 130 they statement that experiment and theory show "remarkable agreement". It is not clear on what basis they say this. Looking at the SI in figure S7, I would say that for the spectra below 1450 cm^{-1} , none of the calculated spectra give better agreement than the other, unless one puts all the weight on the peak just above 900 cm^{-1} . I agree with the authors that the most direct evidence is provided by the spectral region above 1450 cm^{-1} .

We agree with the reviewer that "remarkable agreement" is a purely subjective statement and the quantification of agreement requires the introduction of some similarity factors. We altered the manuscript accordingly.

(3) Regarding this higher frequency region, first of all, on line 135, I believe the authors mean to quote the calculated wavenumber, but the 1505 cm^{-1} that they cite is clearly the experiment and not the theory. Other than this, in the caption of figure S7 where they show the entire spectrum, they state that minor bands above 1600 cm^{-1} may stem from a small fraction of oxocarbenium-type ions. My question is, how do they know that this is only a small fraction? The peak around 1620 cm^{-1} , which seems to be associated with an open structure, is almost as intense as the one at 1560 cm^{-1} . They need to justify the argument that this is only a minor amount. Moreover, they should do this in the main text and not in the caption of a figure in the SI.

We thank the reviewer for noticing this typo.

Also, we fully agree that the relative populations of open- and closed-type structures is non-trivial, as pointed out by reviewers 1 and 3, as well. We now discuss the potential coexistence of these structures for each molecule in the main text. For the mannose-variant, the minor absorption above 1600 cm^{-1} has a comparatively small intensity to the bands at 1503 and 1550 cm^{-1} which indicates, that only a minor fraction of open-type structures is present. For the glucose and galactose variants, on the other hand, we agree that this fraction might not be so small and changed the text accordingly. #

(4) There is mistake on line 164. It should say Figure S8.

Adjusted.

In general, I believe that the authors have probably gotten the assignments right, but they need to tighten the arguments for the comparison between experiment and theory. Otherwise, one could have simply done calculations and drawn conclusions about the structure of the key intermediates from theory alone. Once they tighten this up, I believe that the paper should be suitable for publication.

We thank the reviewer for this very supportive statement and hope that we have now tightened up the assignment sufficiently.

Reviewer #3

The manuscript by Mucha et al presents the structural characterization of an important reaction intermediate in glycosylation via well-resolved, cold ion vibrational spectroscopy. This work is a timely demonstration of the power of cryogenic ion IR spectroscopy in accessing and characterizing directly relevant reaction intermediates. The results should be of interest to a wide audience.

However, an IRMPD study of the same cation intermediate by a Radboud University/FELIX team, just appeared in JACS. [J. Am. Chem. Soc. 2018, 140, 6034–6038] While the data presented in the Mucha et al. manuscript is of much higher quality, the JACS paper finds similar structures and reach similar conclusions. I think that this remove much of the “the first in-depth structural analysis of glycosyl cations” claimed by this manuscript and the novelty necessary for publication in Nature communications.

We thank the referee for this comment and fully agree. The recent work by Boltje et al. that came out shortly after we submitted this manuscript. Needless to say, we have added a paragraph to acknowledge and discuss their work. We also adjusted our statement on the “first” structural characterization of glycosyl cations. However, it is important to point out in this context, that the reported IRMPD spectra are, due to the limited resolution, not sensitive to the ring pucker. As the ring conformation is one of the major determinants for the stereochemical outcome of glycosylation reactions we still think that the here presented data are of substantial novelty and value.

In addition I have the following comments about the manuscript.

1) *The authors should consider linking the structure(s) determined in this study to the reactions involving these intermediates to make the results presented here more impactful to a wider readership. For example, what are the direct implications of the determined structures on the reaction stereochemistry, etc.*

We thank the reviewer for this comment. As described in the introduction of the manuscript, the stereochemical outcome of glycosylation reactions using C2 participating protective groups is well known and promotes the formation of trans-glycosidic linkages. Nevertheless, this question will become very important in future studies involving C4 or C6 participating groups, where there is no apparent correlation between building block design and the stereochemical outcome of the reaction. The diversity of different ring puckers obtained in this study is very interesting, but it does not affect the stereochemical outcome of the reaction. For C4 & C6 participating groups, however, the preference for particular ring conformations likely is a key towards understanding their stereochemical properties.

2) *The crucial identifiers of the cation structure are the C-O/C=O stretch frequencies, and as such, the experimental IR spectra in Fig 2 should extend the frequency range to 1800 cm⁻¹ as shown in the SI. For most species, there are significant activity above 1600 cm⁻¹, possibly indicating presence of minor isomers of oxocarbenium nature. I don't think that this should be hidden in supplementary, as their presence would indicate the fluxional nature of such structures which may influence the reactivity. The ability of the experimental approach to capture higher energy isomers is particularly relevant when dealing with reaction pathways, where the lowest energy structures may actually be less interesting.*

We fully agree with the reviewer's comment that was also raised by the other reviewers. We extended the IR spectra in the main text and now discuss them in more detail. As pointed out correctly, the presence of oxocarbenium structures also highlights the fluxional nature of these structures. Indeed, investigating reaction pathways and their correlation to the respective intermediates is a very important aspect, which, however, would be way beyond the scope of this manuscript. Experiments towards this direction will be performed in the near future.

3) *The fit of the calculated spectra in the finger print region is quite disappointing. In the JACS study, calculations at the B3LYP level appear to produce spectrum that match better with the experiment in the 800-1400 cm⁻¹ region. In any case, this make it difficult to assign a particular ring structure from the spectrum itself, and this assignment is mostly based on the calculated relative energy. If such assignment is to be made, a higher level calculation should be performed to confirm the energetic ordering.*

The great advantage in recording IR spectra using cryogenic methods is that absorption bands are typically very narrow because of the absence of any thermal activation during the measurement. As a result, spectral signatures that would be inaccessible via IRMPD spectroscopy can be clearly resolved. In the light of this, one cannot easily compare the previous JACS publication with the present work. The characteristic bands above 1400 cm⁻¹ are highly indicative of the ion's structure and the high resolution even allows to distinguish between different ring conformations, as shown in Figure S7, S8 and S9.

On the downside, however, the majority of theoretical IR spectra does not match particularly well to these "high-resolution" experimental spectra. This does not necessarily mean that the underlying conformers are different from ones observed experimentally. Instead, the spectral resolution reaches a level where the conventional harmonic approximation and density-functionals used to calculate IR spectra fails to produce a visually appealing match. In our experience, if the experimental and theoretical spectra show broader features, such as in IRMPD spectroscopy, it is easier to achieve a convincing, but yet very subjective agreement.

Finally, to further confirm the identification of lowest-energy structures, we expanded the theoretical calculations to include the energetics of cations at the RI-

MP2 level of theory extrapolated to the complete basis set. The new relative energetics, augmented with harmonic vibrational free energies from PBE0+D3/6-311G+(d,p) calculations, are presented in corrected Figures S7, S8, S9. These energies, however, do not differ substantially from our previous calculations and therefore do not alter the conclusion of the manuscript. #

Minor:

On line 135, the calculated CO stretch of glucose variant is listed as 1505 and 1560 cm⁻¹. But the calculated spectrum in fig. 2A appears to have the CO stretch below 1500 cm⁻¹. Authors may need to double check their values.

We thank reviewer for pointing out the mismatch. The values have been corrected.

REVIEWERS' COMMENTS:

Reviewer #2 (Remarks to the Author):

I was happy with the changes introduced by the authors in response to my remarks. The comments regarding the small percentage of open structures were appropriate and removes any impression that the authors are hiding anything. They also emphasized the the higher frequency region as being the most diagnostic. They also corrected the remaining small mistakes. In light of these changes, I am supportive of its publication in Nature.

Regarding the remarks of the two other reviewers about the recently published JACS paper using IRMPD on room temperature samples, I would not let this impede publication of the Pagel manuscript in Nature. The authors are correct in indicting that the warmer temperatures and resulting broader line shapes make it seem easier to get agreement between experiment and calculations. Moreover, in the IRMPD work, the highly nonlinear nature of the excitation process is well known to distort the intensities, sometimes skewing the conclusions regarding the comparison between experiment and theory. The current work provides a more detailed and stringent test of the glycosal cation structures. The first work published is not always the definitive one.

Reviewer #3 (Remarks to the Author):

The authors have properly addressed my comments and those of the two other reviewers. I recommend publication.